# A combined experimental and computational study of ligand-controlled Chan-Lam coupling of sulfenamides

Kaiming Han[1,2,5], Hong Liu[1,5], Madeline E. Rotella[3,5], Zeyu Xu[1], Lizhi Tao[1], Shufeng Chen[2] ✉, Marisa C. Kozlowski[3] ✉ & Tiezheng Jia[1,4] ✉

The unique features of the sulfenamides' S(II)-N bond lead to interesting stereochemical properties and significant industrial functions. Here we present a chemoselective Chan–Lam coupling of sulfenamides to prepare *N*-arylated sulfenamides. A tridentate pybox ligand governs the chemoselectivity favoring C−N bond formation, and overrides the competitive C-S bond formation by preventing the S,N-bis-chelation of sulfenamides to copper center. The Cu(II)-derived resting state of catalyst is captured by UV-Vis spectra and EPR technique, and the key intermediate is confirmed by the EPR isotope response using ¹⁵N-labeled sulfenamide. A computational mechanistic study reveals that *N*-arylation is both kinetically and thermodynamically favorable, with deprotonation of the sulfenamide nitrogen atom occurring prior to reductive elimination. The origin of ligand-controlled chemoselectivity is explored, with the interaction between the pybox ligand and the sulfenamide substrate controlling the energy of the *S*-arylation and the corresponding product distribution, in agreement with the EPR studies and kinetic results.

Sulfenamides are a class of divalent sulfur-derived scaffolds featuring an S-N bond[1]. For many years, they have been found to be a superior cross-linking agent to elemental sulfur in rubber production, providing greater operational safety and higher cross-linkage yields[2–4]. In the field of medicinal chemistry, sulfenamides have been identified as the active metabolites of proton pump inhibitors in the treatment of acid-related gastrointestinal diseases[5,6]. Other uses of sulfenamides in industry include load-capacity improvers in lubricants[7], wood preservatives[8], and insecticides in agricultures[9]. From the standpoint of organic chemistry, they continue to elicit interest due to their utility as protecting groups in peptide synthesis and their stereochemical properties arising from hindered rotation about the S–N bond[1]. Very recently, we and other groups have exploited sulfenamides as a versatile reagent to prepare other organosulfur pharmacophores with

higher oxidation states, such as S(IV)-derived sulfilimines and S(VI)-derived sulfoximines[10–16]. Notably, C−S bond forming pathways override the alternative C−N bond formation processes either in the presence of transition-metal catalysts or under transition-metal-free conditions.

Considering the unique structural features of sulfenamides, and their widespread applications in academia and industry, much effort has been devoted to their preparation. Conventionally, synthesis of sulfenamides predominantly relies on the construction of S−N bonds via a nucleophilic attack from amines to disulfides, sulfenyl halides or their surrogates (Fig. 1A, a, left)[17]. Moreover, dehydrogenation of amines and thiols under oxidation conditions could lead to the formation of S−N bonds, though condensation of thiols to disulfides often competes (Fig. 1A, a, right)[18]. Alternatively, direct arylation of

[1]Research Center for Chemical Biology and Omics Analysis, Department of Chemistry, Southern University of Science and Technology, 1088 Xueyuan Blvd., Shenzhen, Guangdong, P. R. China. [2]Inner Mongolia Key Laboratory of Fine Organic Synthesis, Department of Chemistry and Chemical Engineering, Inner Mongolia University, Hohhot, P. R. China. [3]Roy and Diana Vagelos Laboratories, Department of Chemistry, University of Pennsylvania, 231 South 34th Street, Philadelphia, Pennsylvania, USA. [4]State Key Laboratory of Elemento-Organic Chemistry, Nankai University, Tianjin, P. R. China. [5]These authors contributed equally: Kaiming Han, Hong Liu, Madeline E. Rotella. ✉e-mail: shufengchen@imu.edu.cn; marisa@sas.upenn.edu; jiatz@sustech.edu.cn

**Fig. 1 | Strategies and Challenges in Synthesis of Sulfenamide. A** Synthetic Approaches to Sulfenamides. **a** Classic Synthetic Strategies to Sulfenamides. **b** This Work: Synthesis of *N*-Arylated Sulfenamides by Chan–Lam Coupling. **B** Conceptual Design and Major Challenges of Our Approach. **a** Conceptual Design of Our Approach. **b** Potential Challenges of Our Approach.

*N*H-sulfenamides represents a straightforward and step-economical pathway to afford *N*-aryl sulfenamides (Fig. 1A, b). Despite this appeal, C–N arylation of *N*H-sulfenamides has not been explored previously, to the best of our knowledge, primarily due to the fact that transition-metal-catalyzed or transition-metal-free functionalization of *N*H-sulfenamides prefers to take place on the sulfur site rather than the nitrogen site[10–16], even though *N*-functionalization is thermo-dynamically favored[11]. Furthermore, the S–N bond of sulfenamides is notoriously labile, as it is prone to hemolysis under thermal or photo-induced conditions[19] or heterolysis in the presence of nucleophiles or electrophiles (Fig. 1B, b)[20]. In addition, sulfur(II)-derived sulfenamides readily convert to the sulfur(IV)-derived sulfinamides or sulfur(VI)-derived sulfonamides upon exposure to oxidants (Fig. 1B, b)[21,22]. To efficiently construct the C–N bond while retaining the fragile S–N bond of sulfenamides, Chan–Lam coupling represents an appealing aryla-tion protocol owing to inexpensive and abundant copper catalysts, mild reaction conditions, broad functional group tolerance, and neu-tral pH conditions[23,24]. Our group has pioneered a copper-catalyzed Chan–Lam type coupling of sulfenamides assisted by an acyl-based protecting group on nitrogen in the absence of an external ligand, and a variety of sulfilimines have been prepared[11]. The chemoselectivity favoring less thermodynamically stable product arising from C–S bond formation was attributed to the bidentate sulfenamide coordination in the transmetalation event. Very recently, we have introduced an enantioselective copper-catalyzed Chan–Lam type *S*-arylation of *N*-aryl sulfenamides with arylboronic acids to furnish chiral sulfilimines, in which the chemoselectivity favoring C–S bond as well as the enan-tioselectivity was steered by a bidentate pyridyl oxazolidine ligand[15]. Subsequently, Yang and coworkers disclosed that transition-metal-free arylation of *N*-aryl sulfenamides with diaryliodonium salts occurs on the sulfur atom even without an acyl directing group on nitrogen[25]. To

achieve C–N bond formation of sulfenamides via a Chan–Lam coupling by overriding the kinetically favored C–S bond formation, we envi-sioned a ligand-controlled Chan–Lam coupling protocol for two major reasons: (1) Considering that most Chan–Lam couplings occur readily without a ligand, generation of a ligand-coordinated copper species would prevent background *S*-arylation by non-ligated copper com-plexes; (2) A multi-dentate ligand could block S,N-bis-chelation of sulfenamides to copper center, thereby forcing only N-binding, which gives rise to a suitable intermediate for C–N bond reductive elimina-tion. Herein, we report a ligand-controlled Chan–Lam coupling of sulfenamides with arylboronic acids to provide facile access to a vari-ety of *N*-arylated sulfenamides with high level of chemoselectivity favoring C–N bond formation over C–S bond formation (Fig. 1B, a).

## Results

### Reaction Optimization

We initiated the investigation by employing *S*-(4-fluorophenyl)-*N*-(*p*-tolyl) thiohydroxylamine (**1a**) and *p*-tolylboronic acid (**2a**) as substrates (Table 1). After a series of optimizations (see Supplementary Tables 1-5), the optimal conditions for Chan–Lam coupling of sulfenamides were determined to be: sulfenamide **1a** as the limiting reagent, boronic acid **2a** (2.0 equiv) as coupling partner, Cu(TFA)$_2$·H$_2$O (10 mol %) as catalyst, pybox **L3**[26] (20 mol %) as ligand, Cy$_2$NMe (1.5 equiv) as base, in MeCN (0.3 M) at room temperature for 24 h under an O$_2$ atmosphere. These conditions provided an 87% assay yield of the desired product **3aa**, with an 84% isolated yield (Table 1, entry 1). Notably, only 5% of the alternate *S*-arylation product was observed. The electron-donating *p*-OMe in ligand **L3** was beneficial to the transformation, as parent **L2** or **L1** bearing an electron-withdrawing *p*-Cl leads to lower yields of **3aa** (entries 2, 3). In control experiments, the copper complex, Cy$_2$NMe, and ligand were found to be essential for the catalytic reaction, as evidenced by no or

**Table 1 | Optimization for Chan–Lam Coupling of Sulfenamide (1a) and *p*-Tolylboronic Acid (2a).ª**

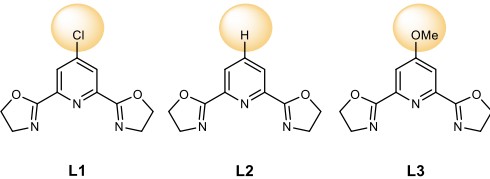

| entry | deviation from standard conditions | assay yield[b]/% |
|---|---|---|
| 1 | none | 87(84[c])/5[d] |
| 2 | **L1** instead of **L3** | 56/11[d] |
| 3 | **L2** instead of **L3** | 72/7[d] |
| 4 | without Cu(TFA)$_2$·H$_2$O | 0 |
| 5 | without Cy$_2$NMe | 0 |
| 6 | without **L3** | trace |
| 7 | under Ar atmosphere instead of O$_2$ | 8 |
| 8 | under air atmosphere instead of O$_2$ | 37 |

**Deviation of Standard Conditions**. ªReaction conditions: **1a** (0.15 mmol), **2a** (2.0 equiv), Cu(TFA)$_2$·H$_2$O (10 mol %), **L3** (20 mol %), Cy$_2$NMe (1.5 equiv) in MeCN (0.5 mL) under O$_2$ at room temperature for 24 h.
[b]Assay yield determined by ¹⁹F NMR analysis of the unpurified reaction mixtures using 0.1 mmol PhCF$_3$ (12.0 μL) as internal standard.
[c]Isolated yield.
[d]Assay yield of *S*-arylation product **3aa'**.

trace of **3aa** in the absence of any of these reagents (Table 1, entries 4-6). Only 8% assay yield of **3aa** was observed in the absence of the external-oxidant O$_2$ (Table 1, entry 7). The theoretical yield of **3aa** under anaerobic condition should be 5%, but the error of 3% could either be attributed to the error of ¹⁹F NMR technique used to determine the assay yield, or caused by the inevitable trace amount of air in the microwave vial. When air was used instead of O$_2$ atmosphere, the assay yield of **3aa** dramatically dropped to 37% (Table 1, entry 8).

## Substrate Scope

With the optimized conditions in hand, we explored the scope of arylboronic acids in the coupling with *N*-phenyl-*S*-(*p*-tolyl)thiohydroxylamine (**1b**) (Fig. 2). Arylboronic acids with electron-donating groups, such as *p*-Me (**2a**), *p*-OMe (**2b**), and *p*-SMe (**2c**) were well tolerated, leading to the formation of desired products (**3ba-3bc**) in 57-87% yields. The coupling reaction proceeded smoothly with arylboronic acids possessing electron-withdrawing groups, including *p*-F (**2d**), *p*-Cl (**2e**), *p*-CF$_3$ (**2 f**) and *p*-CN (**2 g**), affording **3bd-3bg** in 57-81% yields. Sterically hindered 2-tolylboronic acid (**2 h**) was compatible with our protocol, providing **3bh** in 80% yield. Similarly, *meta*-substituted arylboronic acids (**1i** and **1j**) reacted well with **1b** under the standard condition, providing the desired products (**3bi** and **3bj**) in 86% and 72% yield, respectively. Notably, arylboronic acids equipped with different functional groups, including ketone (**1k**), ester (**1 l**), and alkenyl (**1 m**), could be utilized to generate **3bk-3bm** in 54-72% yields. Attesting to the mild reaction conditions and broad functional group tolerance, a myriad of heteroarylboronic acids were found to be compatible under the optimal conditions; the *N*-pyrimidyl (**3bn**), benzofuranyl (**3bo**), thiophenyl (**3bp**), indolyl (**3bq**) and quinolinyl (**3br, 3bs**) sulfenamides were obtained in yields ranging from 40% to 82%, highlighting the expediency of our protocol. The structure of **3bn** was unambiguously assigned by X-ray crystallography (CCDC:

2142998, see Supplementary Fig. 1 for details), which confirms that arylation occurs on nitrogen rather than on sulfur.

Next, the sulfenamide component was evaluated with **2a** (Fig. 2). The aryl group on the sulfur of the sulfenamide was especially insensitive to electronic or steric effects. An *S*-aryl-*N*-phenyl sulfenamide possessing electron-donating *p*-OMe group (**1c**) afforded **3ca** in 56% yield. Electron-withdrawing substituents appended to the *S*-aryl-*N*-phenyl sulfenamide, such as *p*-F (**1a**), *p*-Cl (**1d**) and *p*-NO$_2$ (**1e**) were well tolerated under the standard conditions to deliver the expected products in good yields. Sterically demanding *ortho*-substituents, such as *o*-Me (**1 f**), *o*-OMe (**1 g**), *o*-Cl (**1 h**) and *o*-COOMe (**1i**), were not detrimental, furnishing **3fa-3ia** in a range of 83-90% yields. In particular, the coupling reaction proceeded smoothly with sulfenamide **1j** containing two flanking *o*-Me groups to afford **3ja** in 80% yield. In addition, *S*-aryl sulfenamide (**1k**) with an *m*-Br substituent was viable, providing **3ka** in 80% yield. Our Chan–Lam coupling protocol was also successful with an *S*-heteroaryl sulfenamides to provide **3la** in 54% yield.

The substituent group on the nitrogen of the sulfenamide was also varied. Aryl groups bearing electron-donating substituents such as *p*-Me (**1 m**) or *p*-OMe (**1n**) underwent the coupling reaction with **2a** to afford **3ma** and **3na** in 75% and 62% yield, respectively. Electron-withdrawing substituents, including *p*-F or Cl (**1o, 1p**), were equally successful leading to **3oa** (75%) and **3pa** (75%). Aryl groups on the nitrogen bearing sterically demanding *ortho*-substituents, such as *o*-Me (**1q**) and *o*-F (**1r**), were compatible with our protocol, delivering **3qa** and **3ra** in 61% and 85% yield under slightly modified conditions. Again, a range of functional groups on the *N*-aryl, such as ester (**1 s**), nitrile (**1t**), ketone (**1 u**), sulfonyl (**1 v**) and amide (**1w**), were well tolerated to furnish **3sa-3wa** in 52-65% yields. Even *N*-indolyl-*S*-(*p*-tolyl) sulfenamide (**1x**), an example of an *N*-heteroaryl, could be utilized affording **3xa** in 57% yield under slightly modified conditions. Remarkably, challenging *N*-alkyl sulfenamides, which are usually less stable than their aryl

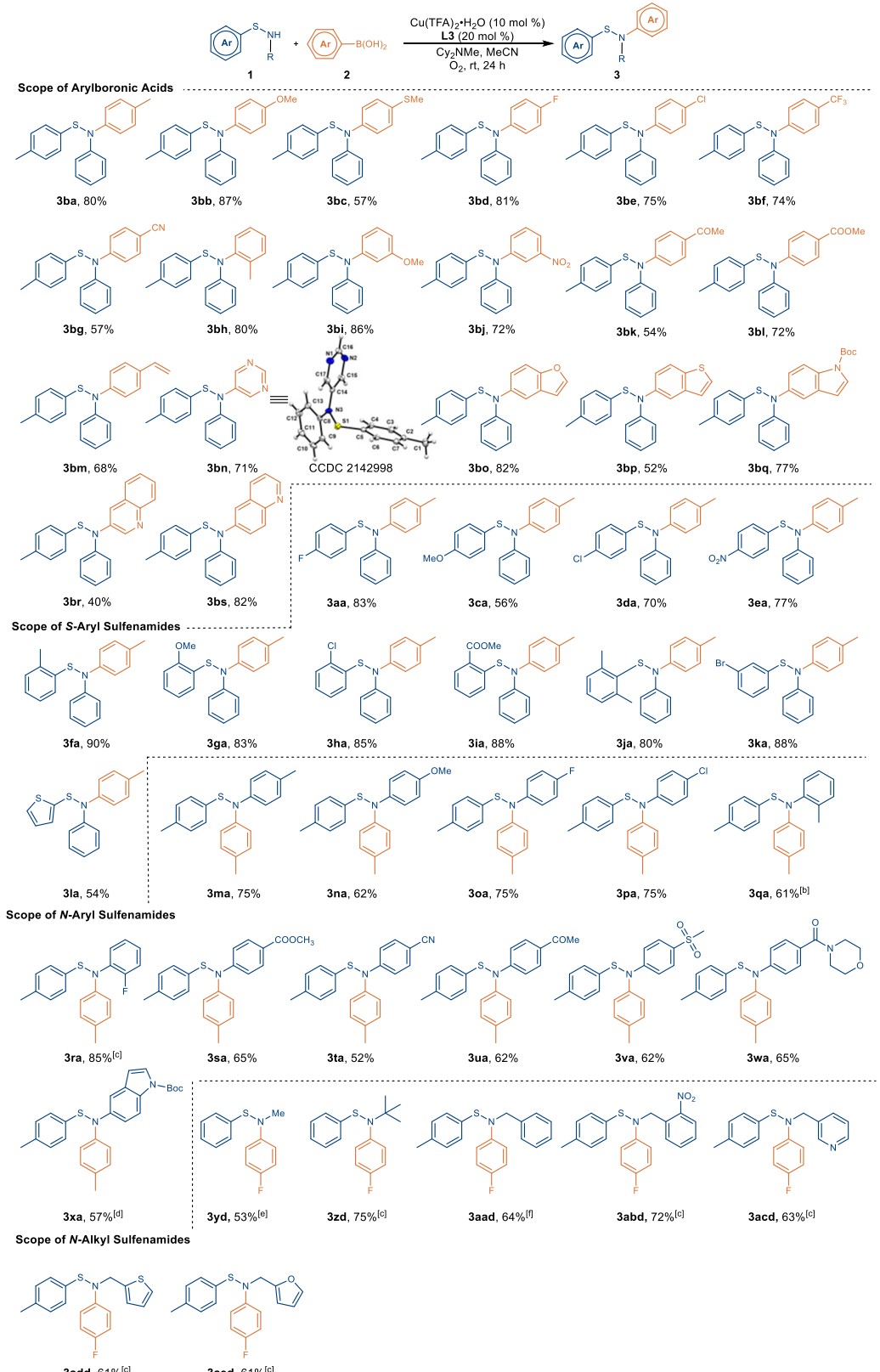

**Fig. 2 | Substrate Scope[a].** [a]Reaction conditions: **1a** (0.15 mmol), **2a** (2.0 equiv), Cu(TFA)₂•H₂O (10 mol %), **L3** (20 mol %), and Cy₂NMe (1.5 equiv) in MeCN (0.5 mL) under O₂ at room temperature for 24 h. [b]35 °C, 24 h. [c]12 h. [d]Cu(TFA)₂•H₂O (15 mol %), **L3** (30 mol %), 12 h. [e]35 °C, 2 h. [f]35 °C, 12 h.

counterparts, were also well-suited under the catalytic conditions. *N*-Methyl sulfenamide **1y** coupled with **2d** to give **3yd** in 53% yield under slightly modified conditions. Substrate bearing bulky *tert*-butyl group on nitrogen (**1z**) could be well tolerated to provide **3zd** in good yield in

12 h. *N*-Benzyl sulfenamide (**1aa**) underwent the coupling reaction smoothly, as evidenced by the formation of **3aad** in 64%. 2-Nitro-benzyl, a photocleavable protecting group was orthogonal to the coupling protocol affording **3abd** in 72% yield, offering a potential site

**Fig. 3 | Synthetic Applications. a** Downstream Transformations of Products. **b** Synthesis of Sulfenamide-Analog of Sulfacetamide. **c** Synthesis of Sulfenamide-Analog of Oxybuprocaine.

for the downstream functionalization. Of note, heteroaryl groups, including 3-pyridyl (**1ac**), 2-thiophenyl (**1ad**), and 2-furanyl (**1ae**), appended on the benzyl position were compatible, extending the substrate generality of our protocol.

## Synthetic Applications

To showcase the potential synthetic applications, further transformations of **3ba** were explored (Fig. 3a). After the Chan–Lam coupling between **1b** and **2a**, direct treatment of the reaction with oxone (2.0 equiv) and diethylamine (40 mol %) in an one-pot fashion led to the formation of sulfinamide **4** in 48% yield (Fig. 3a)[27]. Likewise, adjusting the amount of oxone to 5.0 equivalents could deliver the corresponding sulfonamide **5** in 45% yield (Fig. 3a)[28].

To improve poor aqueous solubility and slow dissolution rates of therapeutics containing acidic N-H bonds, Guarino and coworkers have attempted to use sulfenamide-type analogs as a prodrug strategy[29]. Toward this end, our approach provides a facile tool to directly functionalize sulfenamide-type derivatives of drugs attesting to the mild reaction conditions as well as the broad functional group tolerance, while retaining the labile S-N bond. Sulfacetamide, which is a marketed anti-infective agent in the treatment of conjunctivitis, trachoma and other eye infections[30], could be readily transformed to the sulfenamide-derived compound (**8**) in two steps, 39% overall yield (Fig. 3b). Of note, our Chan–Lam coupling favored the C–N bond formation on the sulfenamide but left secondary amide group intact, highlighting the excellent chemoselectivity. Oxybuprocaine, a short-acting anesthetic for ophthalmology and otorhinolaryngology[31], which was synthesized from commercially available **9** in 4 steps, could be effectively decorated with the *N*-aryl sulfenamide moiety using our Chan–Lam coupling (Fig. 3c).

## Mechanistic Studies

A combined experimental and computational study was performed to gain insight into the mechanism of Chan–Lam coupling (Fig. 4–5). Interestingly, even though the chemoselectivity of Chan–Lam coupling of sulfenamides was affected by the substituents on the pybox ligands (Table 1, entries 1-3), the conversion rates using **L1**–**L3** were similar as illustrated by the kinetic studies (Fig. 4a), suggesting the chemoselectivity favoring C-N bond over C-S bond was not determined by the rate-limiting step.

UV-Vis spectra of sulfenamide **1a**, boronic acid **2a**, and ligand **L3** did not exhibit appreciable absorptions above 400 nm as expected, whereas a mixture of $Cu(TFA)_2 \cdot H_2O$ and **L3** exhibited a strong absorption at 740 nm, in agreement with the d-d transition of Cu(II) complexes (Fig. 4b). Similarly, a characteristic band was observed at 700 nm upon monitoring the reaction of **1a** with **2a** indicating that a copper(II) species serves as the resting state of the catalyst during the catalytic cycle. To identify this copper(II) ($3d^9$, $S=1/2$) species in the reaction, electron paramagnetic resonance (EPR) spectra were recorded (Fig. 4c). In this case, EPR-silent Cu(I)Tc was used with dioxygen gas to mimic the turnover process to form the Cu(II) species in the catalytic cycle. In doing so, two EPR active Cu(II) species were identified. One species (species **Cu-1**) corresponds to Cu(II) bound with **L3** and **2a** in a $d_{x^2-y^2}$ ground state, with an axial **g** tensor of [2.296, 2.054, 2.053] and a hyperfine tensor of $A(^{63}Cu) = [510, 35, 35]$ MHz. In addition, significant superhyperfine coupling with the $^{14}N$ ($I=1$) from the ligand **L3** gives rise to a hyperfine splitting signals at the $g_\perp \sim 2.054$ region. Spectral simulation suggests only two $^{14}N$ nuclei coupled to the electron spin center Cu(II) (see Supplementary Fig. 2-4 for simulation parameters). However, upon further introduction of **1a**, a second species (species **Cu-2**) was observed, which is also in a $d_{x^2-y^2}$ ground state, but with three $^{14}N$ nuclei coupled to the electron spin center

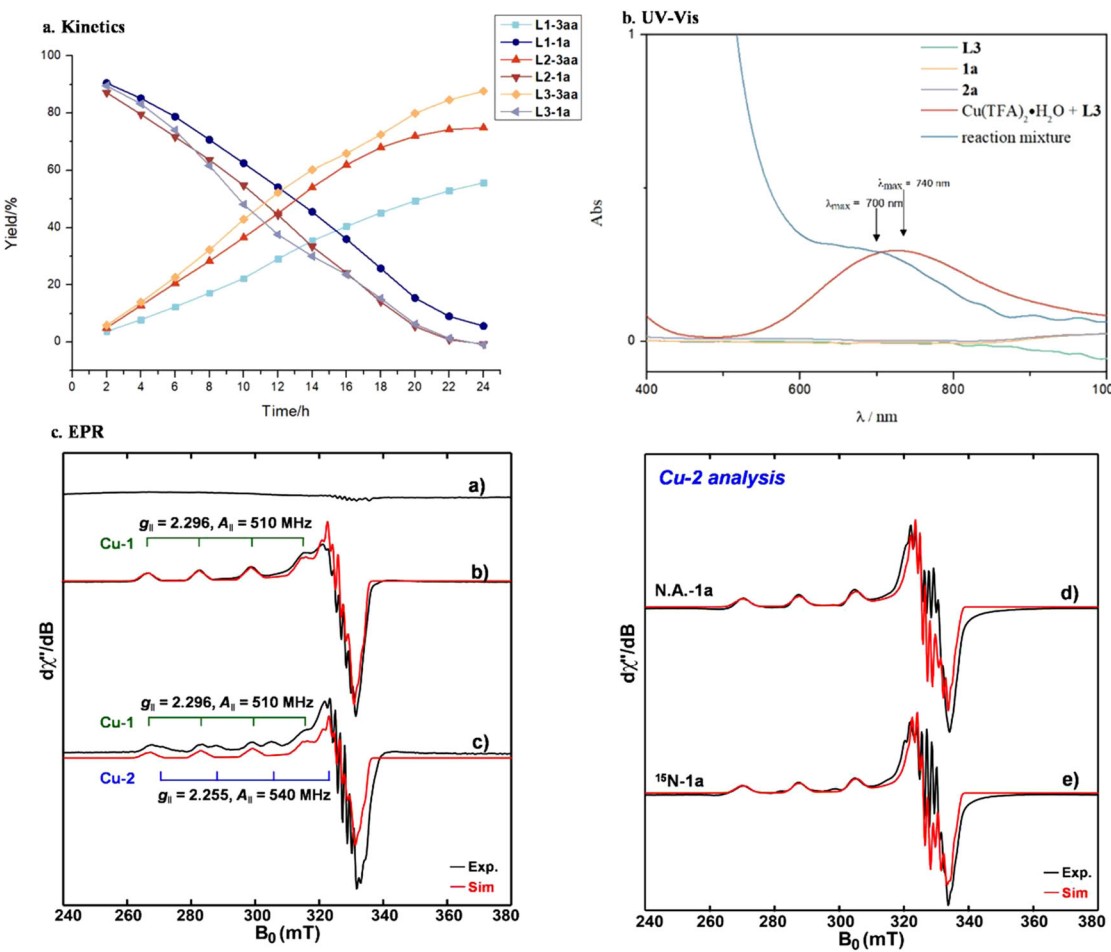

**Fig. 4 | Mechanistic Studies. a** Kinetics. **b** UV–Vis Spectra. **c** EPR Spectra.

Cu(II). When compared to **Cu-1**, this result indicates that the third $^{14}$N of species **Cu-2** most likely arises from **1a** binding to Cu(II) via nitrogen. This assignment was confirmed by the EPR isotope response when using $^{15}$N labeled **1a** (Fig. 4c, Supplementary Fig. 5-6).

To understand the factors affecting chemoselectivity and canonical steps, the mechanism for this transformation with **L2** was probed using density functional theory (DFT) [UM06/6-311 + + G(d,p)-SDD(Cu)-CPCM(DME)//UB3LYP-D3/6-31 G(d)-SDD(Cu)[32–40], see Supplementary Fig. 7-49 for full computational details]. In Fig. 5a, four- and five-coordinate cationic copper complexes were explored computationally based on X-ray crystallographic evidence for similar structures[41–43]. Initially, the arylboronic acid is proposed to form the pre-reacting complex **16'** (Fig. 5a, uphill in energy by 3.5 kcal/mol) from the starting cationic Cu$^{II}$ complex **16**. This complex then undergoes transmetalation (via **[16-17]**, 9.9 kcal/mol) to form intermediate **17** (−10.3 kcal/mol) observed by EPR spectroscopy (species **Cu-1** in Fig. 4c). The boronic acid disrupts the tridentate binding of the pybox ligand via H-bonding with one nitrogen atom of the oxazoline ring, accounting for the observed binding of only two nitrogen atoms in the EPR spectra. Next, sulfenamide exchanges with boronic acid to form **18** (−12.9 kcal/mol) which also has only two nitrogen atoms of the pybox ligand bound to the copper center, consistent with EPR studies (species **Cu-2** in Fig. 4c) (for a comparison of the energetics of **18** and its conformer with different methods, see Supplementary Fig. 7).

Intermediate **18** then undergoes deprotonation with NMe$_3$ (as a simplified model for Cy$_2$NEt) at the nitrogen atom of the sulfenamide via **[18-19]** (overall energetic span of 13.1 kcal/mol) to form intermediate **19**. Cu$^{II}$ species **19** undergoes disproportionation with Cu$^{II}$ intermediate **16** to generate Cu$^{III}$ intermediate **20** and Cu$^{I}$ hydroxide[24]

(transition state not located). Following disproportionation, the selectivity-determining arylation occurs. On the other hand, intermediate **20**, which has the sulfur atom of the sulfenamide bound to the copper center, can undergo reductive elimination via **[20-21]** (overall span of 5.0 kcal/mol from **19**) to generate S-aryl product **21**. Alternatively, the sulfenamide can coordinate via the nitrogen atom as in Cu$^{III}$ intermediate **20'**. Intermediate **20'** will then undergo facile reductive elimination via **[20'-21']** (span of 0.4 kcal/mol) to yield the experimentally observed and thermodynamically favored N-aryl product **21'** (downhill in energy by 61.5 kcal/mol) and regenerate Cu$^{I}$. The Cu$^{I}$ species is oxidized by oxygen to Cu$^{II}$ to restart the catalytic cycle[24], as shown in Fig. 5b.

Next, we explored the nature of the ligand-controlled chemoselectivity by comparing the experimental results with **L1**, **L2**, and **L3** to our computational results. The potential energy surface for the formation of S-aryl and N-aryl products from intermediate **18** with these ligands is given in Supplementary Fig. 8. While the spans for N-arylation for **L1**, **L2**, and **L3** are fairly similar (see Supplementary Fig. 9-10), the S-arylation for **L3** has the highest span followed by **L2** and **L1**. Notably, ligand **L3**, which experimentally gives the best chemoselectivity for the N-arylation product, has the largest difference between energy spans for S-arylation and N-arylation, supporting our mechanistic proposal.

To understand the origin of ligand-controlled chemoselectivity, interaction/distortion analysis was performed on the S-arylation and N-arylation transition states as described by Houk and Bickelhaupt[44] (see Supplementary Fig. 11). To estimate the electronic energy of the transition states, a comparison analysis of the favorable interaction energy between CuL and aryl substrate with the disfavorable distortion energy

**Fig. 5 | Computational Mechanistic Studies. a** DFT Computational Study: Formation of *N*-Arylation Product 21′. All Free Energies Were Computed Using UM06/6-311 + + G(d,p)-SDD(Cu)-CPCM(DME)//UB3LYP-D3/6-31 G(d)-SDD(Cu). **b** Proposed Catalytic Cycle for Chan–Lam Coupling of Sulfenamides.

from the intermediate into the transition state geometries was performed. For *S*-arylation, the interaction energy appears to control the energy of the transition states. Specifically, *S*-arylation with **L1** has the lowest overall energy and the most favorable interaction energy,

followed by **L2** and **L3**. This favorable interaction energy for *S*-arylation with **L1** leads to the observed lower ratio of N:S product experimentally. Plots of the noncovalent interactions (NCI) show the slightly larger interaction between the aryl of the sulfenamide and the ligand in

**L1** compared with **L2** or **L3** (Supplementary Fig. 12). On the other hand, the interaction energies between the **L**Cu fragment and the sulfenamide in the *N*-arylation are fairly similar for all three ligands. Instead, it is the distortion energy which appears to control the relative energies of the transition states, with **L3** having the largest distortion energy, corresponding with its highest energy. This distortion can be observed in the bond lengths of the forming C-S bond (Supplementary Fig. 13), where **L3** has the shortest C-S bond. It follows that this transition state is the latest of the three, which leads to greater distortion from the initial ground state.

From the combined experimental results and computational studies, a plausible catalytic cycle was proposed (Fig. 5b). Oxidation of Cu$^I$ species **A** leads to the formation of Cu$^{II}$ complex **B** under exposure to dioxygen gas, followed by transmetalation with arylboronic acid **2** to yield arylated Cu$^{II}$ complex **C**. Then, sulfenamide **1** binds to copper center via the nitrogen atom, as supported by the EPR study. Next, deprotonation occurs, followed by disproportionation to furnish a Cu$^{III}$ species (**F**) along with a Cu$^I$ byproduct. Ultimately, **F** undergoes reductive elimination to afford the desired sulfenamide **3**, and **A** was formed to close the catalytic cycle.

In this work, we have developed a copper-catalyzed Chan–Lam coupling of sulfenamides with arylboronic acids to afford diverse *N*-aryl sulfenamides in a direct and step-economical fashion. Sulfenamide-type derivatives of two marketed therapeutics have been directly functionalized using our protocol, highlighting its synthetic potential in medicinal chemistry. The chemoselectivity favoring C−N bond formation over C-S bond formation is steered by a tridentate pybox ligand, which dictates mono-coordination of the sulfenamide to the copper center via nitrogen, and facilitates the subsequent reductive elimination of C−N bond. The key copper species in the catalytic cycle are spectroscopically investigated, and mono-coordination of sulfenamide to copper via the nitrogen atom is characterized by EPR and verified by the EPR isotope response using $^{15}$N-labeled sulfenamide. DFT studies show that the reaction proceeds via transmetalation of the arylboronic acid followed by coordination of the sulfenamide substrate, deprotonation of the nitrogen atom, disproportionation, and reductive elimination. With this ligand, *N*-arylation is kinetically and thermodynamically favorable. The exact nature of the ligand determines the degree of *S*-arylation, with the electron-rich **L3** having the highest N:S product ratio. Our work provides strong evidence that Chan–Lam coupling, besides its mild conditions and broad functional group tolerance, could be manipulated by ligands as a precise synthetic tool, which is currently under investigation in our laboratories.

## Methods

### General Procedure for Catalysis

To an oven-dried microwave vial equipped with a stir bar was added Cu(TFA)$_2$•H$_2$O (4.5 mg, 10 mol %), **L3** (7.4 mg, 20 mol %) *S*-(4-fluorophenyl)-*N*-(*p*-tolyl)thiohydroxylamine (**1a**) (35.0 mg, 0.15 mmol) and *p*-tolylboronic acid (**2a**) (40.8 mg, 0.3 mmol). Cy$_2$NMe (48.2 μL, 0.23 mmol) and MeCN (0.5 mL) were added via syringe under an air atmosphere. The vial was sealed with a septum, and refilled from a dioxygen balloon for 3 min. The solution was then stirred at room temperature for 24 h under an O$_2$ atmosphere. Upon completion of the reaction, the solvent was removed under vacuum to give a residue, which was further purified by flash chromatography to give the pure product.

## Data availability

Detailed experimental procedures, characterization data, NMR spectra of new compounds, detailed computational results, and calculated structures are available within Supplementary Information. The X-ray crystallographic coordinates for structures reported in this study have been deposited at the Cambridge Crystallographic Data Centre (CCDC), under deposition numbers CCDC 2142998 (for **3bn**). These data can be obtained free of charge from The Cambridge Crystallographic Data Center via www.ccdc.cam.ac.uk/data_request/cif. Any further relevant data are available from the authors upon request.

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

## Acknowledgements

T.J. thanks the National Natural Science Foundation of China (U23A20528), Guangdong Basic and Applied Basic Research Foundation (2021B1515120046, 2022B1515120075), and the Science and Technology Innovation Commission of Shenzhen Municipality (JCYJ20220818101404010, 20220815113214003) for financial support. M.C.K. thanks the NIH (R35 GM131902) for financial support and ACCESS (TG-CHE120052) for computational support. S.C. thanks the National Natural Science Foundation of China (22061031) for financial support. We are also very grateful to Dr. Yang Yu and Dr. Xiaoyong Chang (both at SUSTech) for HRMS and X-ray crystallography, respectively. We acknowledge the assistance of SUSTech Core Research Facilities.

## Author contributions

T.J. conceived and directed the project. K.H., H.L., and Z.X. performed the experiments. M.C.K. directed the part of the computational study. M.E.R. carried out the computational study. L.T. directed the EPR study. T.J., M.C.K., L.T., and S.C. analyzed the data. T.J., M.C.K., and L.T. wrote the manuscript. All authors approved the submission of the manuscript. K.H., H.L., and M.E.R. contributed equally.

## Competing interests

The authors declare no competing interests.
