## [Peer Review File · Nature Communications]

A Combined Experimental and Computational Study of Ligand-Controlled Chan-Lam Coupling of SulfenamidesREVIEWER COMMENTS

Reviewer #1 (Remarks to the Author):

Recently, transformation of sulfenamides has been a very active research topic, but it is mainly focused on the conversion of S(II)-based sulfenamides to S(IV)-derived sulfilimines via S-arylation or alkylation. In this manuscript by Jia, Kozlowski, Chen and coworkers, a highly chemoselective ligand-controlled C-N Chan-Lam coupling of sulfenamides with arylboronic acids to N-arylated sulfenamides is described. In general, Chan-Lam coupling can occur without the external ligands, so the chemoselective ligand-controlled Chan-Lam coupling is very rare. In this regard, the unprecedented and practical method reported herein represents a significant advance in the field, and fits the high standard of Nature Communication. More importantly, the authors also conducted in-depth mechanistic studies to unveil the origin of chemoselectivity and to establish the reaction profile, using UV-Vis spectra, EPR technique and DFT calculations. It was concluded that the effect of tridentate pybox ligand dictated mono-coordination of the sulfenamide to the copper center is the key for C-N bond formation over C-S bond formation. I recommend its acceptance by Nature Communication after addressing the following minor issues.

- (1) In Table 1, the copper catalyst and base in the chemical formula should be clearly defined.
- (2) In Table S4, the yield of sulfilimine 3aa' should be provided to better align with the title of the manuscript.
- (3) Reference 44 lacks page numbers.

Reviewer #2 (Remarks to the Author):

The manuscript presents an investigation combining experimental and computational methods on the Chan-Lam coupling of sulfonamides. While the reported reactions are important and interesting, there are several significant issues that need to be addressed.

One of the key aspects is the requirement for dioxygen as the oxidant to facilitate the catalytic process. In the absence of dioxygen, it is evident that the Cu(II) catalyst serves as the oxidant. In this case, two equivalents of Cu(II) are needed in order to convert one equivalent of sulfonamide to the product. However, it is puzzling to observe in Table 1, entry 7, that using only 10 mol% of Cu(II) produces 10 mol% of the product. It is difficult to comprehend this result unless the authors can experimentally demonstrate and confirm the reduction of Cu(II) to Cu(0) during the reaction.

The DFT calculations provided in the manuscript are incomplete, and the corresponding discussion is quite superficial. This makes it challenging for readers to discern the insights gained from these calculations. Specifically, it remains unclear how the disproportionation reaction occurs and whether the associated barrier is sufficiently high to become rate-determining. Additionally, the manuscript lacks a clear explanation of how 21' reverts back to the active species 16 to complete a catalytic cycle. The energy profile shown in Scheme 3 does not meet the quality standards expected for publication in Nature Communications.

Furthermore, Scheme 1 appears to be disconnected from the text, as it is not referenced anywhere in the manuscript.

Considering the aforementioned concerns, I am hesitant to support the publication of this manuscript without significant revisions and clarifications.

Reviewer #3 (Remarks to the Author):

Sulfenamides are a class of interesting and synthetically useful platform molecules. Previous works mainly focus on the development of sulfenamides as a versatile reagent to prepare organosulfur pharmacophores with higher oxidation states, such as sulfilimines and sulfoximines. In previous works C-S bond forming pathways override the alternative C-N bond formation processes either in the presence of transition metal catalysts or with bases as the promoters. In this work Jia, Kozlowski, Chen and co-workers report a highly chemo-selective copper-catalyzed ligand-controlled Chan-Lam coupling of sulfenamides with the reaction site on the nitrogen atom, which provided a straightforward and step economical route to N-arylated sulfenamides. UV Vis spectra and EPR technique demonstrate that sulfenamide binds to copper center via nitrogen. DFT mechanistic studies showed that N-arylation is both kinetically and thermodynamically favorable, which caused by the interaction between the pybox ligand and the sulfenamide substrate. Additionally, the authors demonstrated the utilities by using their method to derivatize two sulfenamide type analogues of marketed therapeutics. Overall, this is an interesting work and the paper is well organized, thus I recommend publication of this work on nature communications after the following revisions:

- 1) For the N-aryl sulfenamide substrate, what about the results when a substituent was placed at the ortho-position of the N-aryl group?
- 2) In the abstract, the author mentioned that the sulfenamides' S(II)-N bond include a chiral axis. This reviewer thinks the presentation is not so accurate. Since there are no hindered groups around the axis for reported sulfenamides, the configuration of the sulfenamides is labile.

REVIEWER COMMENTS

Reviewer #1 (Remarks to the Author):

Recently, transformation of sulfenamides has been a very active research topic, but it is mainly focused on the conversion of S(II)-based sulfenamides to S(IV)-derived sulfilimines via S-arylation or alkylation. In this manuscript by Jia, Kozłowski, Chen and coworkers, a highly chemoselective ligand-controlled C-N Chan-Lam coupling of sulfenamides with arylboronic acids to N-arylated sulfenamides is described. In general, Chan-Lam coupling can occur without the external ligands, so the chemoselective ligand-controlled Chan-Lam coupling is very rare. In this regard, the unprecedented and practical method reported herein represents a significant advance in the field, and fits the high standard of Nature Communication. More importantly, the authors also conducted in-depth mechanistic studies to unveil the origin of chemoselectivity and to establish the reaction profile, using UV-Vis spectra, EPR technique and DFT calculations. It was concluded that the effect of tridentate pybox ligand dictated mono-coordination of the sulfenamide to the copper center is the key for C-N bond formation over C-S bond formation. I recommend its acceptance by Nature Communication after addressing the following minor issues.

(1) In Table 1, the copper catalyst and base in the chemical formula should be clearly defined.

Our reply: Following Reviewer 1's suggestion, we have clearly defined the copper catalyst, ligand, and base as $\text{Cu}(\text{TFA})_2 \cdot \text{H}_2\text{O}$, ligand **L3** and Cy_2NMe in Table 1.

(2) In Table S4, the yield of sulfilimine **3aa'** should be provided to better align with the title of the manuscript.

Our reply: Following Reviewer 1's suggestion, we have provided the corresponding assay yields of **3aa'** in Table S4.

(3) Reference 44 lacks page numbers.

Our reply: Thanks for Reviewer 1's careful reading, and we have added page numbers to reference 44 (now as reference 43).

Reviewer #2 (Remarks to the Author):

The manuscript presents an investigation combining experimental and computational methods on the Chan-Lam coupling of sulfenamides. While the reported reactions are important and interesting, there are several significant issues that need to be addressed.

(1) One of the key aspects is the requirement for dioxygen as the oxidant to facilitate the

catalytic process. In the absence of dioxygen, it is evident that the Cu(II) catalyst serves as the oxidant. In this case, two equivalents of Cu(II) are needed in order to convert one equivalent of sulfonamide to the product. However, it is puzzling to observe in Table 1, entry 7, that using only 10 mol% of Cu(II) produces 10 mol% of the product. It is difficult to comprehend this result unless the authors can experimentally demonstrate and confirm the reduction of Cu(II) to Cu(0) during the reaction.

Our reply: We repeated the control experiment of 10 mol % Cu(II) catalyst under the anaerobic conditions, and 8% assay yield of **3aa** has been delivered, which has been updated in the revised manuscript. We agree with Reviewer 2 on the point that the theoretical yield of **3aa** under anaerobic condition should be 5%, but believe this error (3%) could be attributed to the error of ¹⁹F NMR technique used to determine the assay yield. In addition, it was inevitable that trace amount of air remained in microwave vial. By Ideal Gas Law:

$$PV = nRT$$

the P(O₂) of O₂ residue needed in vial to cause 3 mol % Cu^I oxidized was calculated (volume of microwave vial is 10.0 mL), considering 1 M of O₂ could donate 4 M of electrons but 1 M of Cu(I) to Cu(II) only needs 1 M of electrons:

$$P(\text{O}_2) \times 10.0 \times 10^{-6} \text{ m}^3 = (3\% \times 0.1 \times 10^{-3} \text{ mol} \times 1/4) \times 8.314 \text{ J}/(\text{mol} \cdot \text{K}) \times 298 \text{ K}$$

$$P(\text{O}_2) = 186 \text{ pa}$$

As the reaction performed under 1 atm,

$$E_r = 186 \text{ pa}/101000 \text{ pa} = 0.18\%$$

So, only 0.18% O₂ is needed to oxidize 3 mol % Cu^I to Cu^{II}.

(2) The DFT calculations provided in the manuscript are incomplete, and the corresponding discussion is quite superficial. This makes it challenging for readers to discern the insights gained from these calculations. Specifically, it remains unclear how the disproportionation reaction occurs and whether the associated barrier is sufficiently high to become rate-determining.

Our reply: Both the *S*- and *N*-arylation pathways must undergo disproportionation prior to product formation per prior mechanistic work by Watson and others. Despite our best efforts to locate the disproportionation transition state in which two Cu^{II} species come together to form a Cu^I and a Cu^{III}, we were unable to find the transition state. However, the selectivity-determining step occurs *after* the disproportionation. As a result, whether the disproportionation reaction is rate-limiting does not affect the selectivity which is what is being studied with the calculations. Additional sentences have been added to the page-long discussion of the calculated catalytic cycle.

(3) Additionally, the manuscript lacks a clear explanation of how 21' reverts back to the active species 16 to complete a catalytic cycle. The energy profile shown in Scheme 3 does not meet the quality standards expected for publication in Nature Communications.

Our reply: Following Reviewer 2's suggestion, we have improved the quality and increased the size of the energy profile in Scheme 3 to ensure it meets the quality standards expected for publication in *Nature Communications*. As a result, the original Scheme 3 was split into Scheme 3 and Scheme 4 in the revised manuscript. The computational portion of Scheme 3 is given below. Furthermore, the Scheme 4a now includes the oxidation of Cu^I to Cu^{II} to show how the catalytic cycle restarts. This is also shown in Scheme 4b.

(4) Furthermore, Scheme 1 appears to be disconnected from the text, as it is not referenced anywhere in the manuscript.

Our reply: We really appreciate Reviewer 2's careful reading and have added corresponding citations of Scheme 1 in page 2, which are highlighted in yellow.

Considering the aforementioned concerns, I am hesitant to support the publication of

this manuscript without significant revisions and clarifications.

Reviewer #3 (Remarks to the Author):

Sulfenamides are a class of interesting and synthetically useful platform molecules. Previous works mainly focus on the development of sulfenamides as a versatile reagent to prepare organosulfur pharmacophores with higher oxidation states, such as sulfilimines and sulfoximines. In previous works C-S bond forming pathways override the alternative C-N bond formation processes either in the presence of transition metal catalysts or with bases as the promoters. In this work Jia, Kozłowski, Chen and co-workers report a highly chemo-selective copper-catalyzed ligand-controlled Chan-Lam coupling of sulfenamides with the reaction site on the nitrogen atom, which provided a straightforward and step economical route to N-arylated sulfenamides. UV-Vis spectra and EPR technique demonstrate that sulfenamide binds to copper center via nitrogen. DFT mechanistic studies showed that N-arylation is both kinetically and thermodynamically favorable, which caused by the interaction between the pybox ligand and the sulfenamide substrate. Additionally, the authors demonstrated the utilities by using their method to derivatize two sulfenamide type analogues of marketed therapeutics. Overall, this is an interesting work and the paper is well organized, thus I recommend publication of this work on nature communications after the following revisions:

1) For the N-aryl sulfenamide substrate, what about the results when a substituent was placed at the ortho-position of the N-aryl group?

Our reply: Sulfenamides bearing *ortho* substituent in *N*-aryl groups, such as *o*-Me (**1q**) and *o*-F (**1r**), underwent the coupling with **2a** smoothly to afford the corresponding products in 61% and 85% yields under slightly modified conditions. These results have been updated into Table 2 and highlighted in yellow.

Reaction conditions: **1a** (0.15 mmol), **2a** (2.0 equiv), Cu(TFA)₂·H₂O (10 mol %), **L3** (20 mol %), and Cy₂NMe (1.5 equiv) in MeCN (0.5 mL) under O₂ at room temperature for 12 h. *35 °C, 24 h.

2) In the abstract, the author mentioned that the sulfenamides' S(II)-N bond include a chiral axis. This reviewer thinks the presentation is not so accurate. Since there are no hindered groups around the axis for reported sulfenamides, the configuration of the

sulfenamides is labile.

Our reply: We agree with Reviewer 3 on the point that the configuration of the sulfenamides is very labile. However, in precedent literatures (*J. Am. Chem. Soc.* **1969**, *91*, 6677-6683; *J. Am. Chem. Soc.* **1979**, *101*, 5890-5895), it has been observed that the S-N bond of sulfenamides ($R^1SNR^2R^3$) is a chiral axis when R^1 is 2,4-dinitrophenyl and R^2 , R^3 are large substituents based on NMR studies.

REVIEWERS' COMMENTS

Reviewer #2 (Remarks to the Author):

In this revised manuscript, the authors have addressed the majority of my comments.

I appreciate the revision made in response to my concern about the 10% yield in the absence of di-oxygen, which has been reduced to 8%. The provided explanation is satisfactory. However, I would suggest that the authors include a brief comment in the main text to enhance clarity for the readers.

The quality of the energy profile (Scheme 4 and Figure S8) has significantly improved compared to the original version. Nevertheless, some of the thick bars appear distorted. I understand that this distortion may be attributed to the drawing software when connecting the thick bars with additional dashed lines. However, it is worth noting that many high-quality figures presented in the literature, which were also created using the same software, do not exhibit this issue. Therefore, I believe there must be a solution to rectify this problem.

Reviewer #3 (Remarks to the Author):

The authors have addressed the reviewer's concerns, thus I recommend publication of this work on nature communications in this version.

REVIEWER COMMENTS

Reviewer #2 (Remarks to the Author):

(1) I appreciate the revision made in response to my concern about the 10% yield in the absence of di-oxygen, which has been reduced to 8%. The provided explanation is satisfactory. However, I would suggest that the authors include a brief comment in the main text to enhance clarity for the readers.

Our reply: Following Reviewer 2's suggestion, a brief comment has been added in the paragraph of "Reaction Optimization" on page 3 and highlighted in yellow.

(2) The quality of the energy profile (Scheme 4 and Figure S8) has significantly improved compared to the original version. Nevertheless, some of the thick bars appear distorted. I understand that this distortion may be attributed to the drawing software when connecting the thick bars with additional dashed lines. However, it is worth noting that many high-quality figures presented in the literature, which were also created using the same software, do not exhibit this issue. Therefore, I believe there must be a solution to rectify this problem.

Our reply: We appreciate Reviewer 2's careful reading and have modified all the thick bars in Scheme 4 and Supplementary Figure 8.

Reviewer #3 (Remarks to the Author):

The authors have addressed the reviewer's concerns, thus I recommend publication of this work on nature communications in this version.

Our reply: We really appreciate Reviewer 3's support.